# mReFinED: An Efficient End-to-End Multilingual Entity Linking System

**Peerat Limkonchotiwat**[1],[*] **Weiwei Cheng**[2]**, Christos Christodoulopoulos**[2],
**Amir Saffari**[2]**, Jens Lehmann**[2]

[1]School of Information Science and Technology, VISTEC, Thailand
[2]Amazon
peerat.l_s19@vistec.ac.th
{wwcheng, chrchrs, amsafari, jlehmnn}@amazon.com

## Abstract

End-to-end multilingual entity linking (MEL) is concerned with identifying multilingual entity mentions and their corresponding entity IDs in a knowledge base. Prior efforts assume that entity mentions are given and skip the entity mention detection step due to a lack of high-quality multilingual training corpora. To overcome this limitation, we propose mReFinED, the first end-to-end MEL model. Additionally, we propose a bootstrapping mention detection framework that enhances the quality of training corpora. Our experimental results demonstrated that mReFinED outperformed the best existing work in the end-to-end MEL task while being 44 times faster.

## 1 Introduction

End-to-end entity linking (EL) is the task of identifying entity mentions within a given text and mapping them to the corresponding entity in a knowledge base. End-to-end EL plays a crucial role in various NLP tasks, such as question answering (Nie et al., 2019; Asai et al., 2020; Hu et al., 2022) and information retrieval (Zhang et al., 2022).

To clarify the terminology used in this paper and previous work, it should be noted that when referring to EL in previous work, we are specifically referring to entity disambiguation (ED) where the entity mentions are given. We will only refer to our definition of EL (mention detection with ED) using the full name "end-to-end EL". Existing EL research has extended models to support over 100 languages in a single model using Wikipedia as the training corpus. We call this task multilingual entity linking (MEL). Recent work proposed MEL frameworks by minimizing discrepancy between mention and entity description representation (Botha et al., 2020) or the same mention but in different contexts (FitzGerald et al., 2021) based on bi-encoder pre-trained language

models (Devlin et al., 2019). An alternative method is predicting the target entity's Wikipedia title in an auto-regressive manner (Cao et al., 2022) by extending the sequence-to-sequence pipeline of Cao et al. (2021). However, none of the existing works perform end-to-end MEL because of a lack of high-quality multilingual entity mention training resources. For example, we found that Wikipedia suffers from an unlabelled entity mention problem, i.e. not all entity mentions have hyperlink markups to train a reliable mention detection model. Thus, devising an end-to-end MEL system remains a challenging task. In this paper, we propose the first end-to-end MEL system. To address the unlabelled mention problem in end-to-end MEL, we propose a bootstrapping mention detection (MD) framework. Our framework leverages an existing multilingual MD model to create a bootstrapped dataset, which we use to train a new mention detection model for annotating unlabelled mentions in Wikipedia. The framework provides an improvement for detecting named and non-named entities in Wikipedia, compared to previous multilingual MD approaches (Honnibal et al., 2020; Hu et al., 2020; Tedeschi et al., 2021). To construct the end-to-end MEL system, we extend ReFinED (Ayoola et al., 2022) since it is comparable to the state-of-the-art (SOTA) models in the English end-to-end EL setting, and significantly faster than any other methods to date. We call this new model *mReFinED*. Our code is released at: https://github.com/amazon-science/ReFinED/tree/mrefined.

To demonstrate mReFinED's effectiveness, we compare it with SOTA MEL (Cao et al., 2022) on the end-to-end MEL task across two datasets, Mewsli-9 (Botha et al., 2020) and TR2016[hard] (Tsai and Roth, 2016). Experimental results show that mReFinED outperforms a two-stage model (combining SOTA MD and MEL models) on both datasets. Moreover, mReFinED's inference speed

---
* Work conducted during Research Internship at Amazon.

is 44 times faster than SOTA MEL.

Our contributions are as follows: We propose the first end-to-end MEL in a single model by extending ReFinED to multilingual ReFinED. In addition, we propose a bootstrapping mention detection framework to solve the unlabelled mention problem in end-to-end MEL.

## 2  Methodology

**Overview**. We first fine-tune a mention detection (MD) model – based on a multilingual pre-trained language model (PLM) – with our bootstrapping MD framework, as shown in Figure 1. We then use the bootstrapped MD model to annotate unlabelled mentions in Wikipedia. Finally, we use the data bootstrapped to train mReFinED in a multi-task manner (Ayoola et al., 2022), which includes mention detection, entity type prediction, entity description, and entity disambiguation.

### 2.1  Bootstrapping Mention Detection

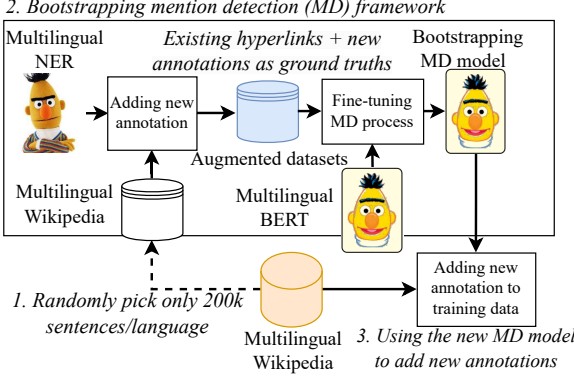

Figure 1: The bootstrapping MD framework's overview.

As shown in Figure 1, we employ an existing multilingual named entity recognition (NER) model to annotate unlabelled mentions in Wikipedia corpora. Our framework allows various choices of existing NER models (Honnibal et al., 2020; Hu et al., 2020; Tedeschi et al., 2021) without any constraints. Based on our MD experiments in section 3.3, we adopted the XTREME NER model (fine-tuned on 40 languages) since it supports every language in both MEL datasets.

We then train a bootstrapped multilingual MD model using the same PLM as previous MD works (mBERT; Devlin et al., 2019) with the newly annotated mentions and existing markups in Wikipedia. For simplicity, we train the bootstrapped MD model similarly to BERT's token classification fine-tuning (using the BIO tagging format; Ramshaw and Marcus, 1995). When there are possibilities of overlapping entities, we only use the longest entity mention. In addition, we found that using only 200k sentences per language as training data is enough for formulating the bootstrapped MD (see Appendix A.3 for a further details). Finally, we use the new MD model to annotate unlabelled mentions in Wikipedia. The main advantage of using the new MD model over the XTREME model is its ability to detect both named and common noun entities. This is because the new MD model learned named entities from the XTREME model's annotation and common noun entities from existing markups in Wikipedia, which is useful since most entities in current MEL datasets are common noun entities. However, the XTREME model is trained only on NER datasets and using it on MEL datasets would harm performance.

### 2.2  Task Definition: end-to-end MEL

Given a sequence of words in the document $W = \{w_1, w_2, ..., w_{|W|}\}$, where the document can be written in multiple languages. We identify entity mentions within the document $M = \{m_1, m_2, ...m_{|M|}\}$ and map each one to the corresponding entity $E = \{e_1, e_2, ..., e_{|E|}\}$ in knowledge base (KB).

### 2.3  Mention Detection

We use mBERT as our mention encoder to encode the words $w_i$, with the output from the last layer serving as the contextualized word representations $\mathbf{h_i}$. We add a linear layer to our mention encoder to train token classification (BIO tagging format) from $\mathbf{h_i}$ using cross-entropy $\mathcal{L}_{\text{MD}}$ to the gold label. Then, we obtain mention representations $\mathbf{m_i}$ for each $m_i$ by average pooling $\mathbf{h_i}$ of the entity mention tokens. All words $W$ are encoded in a single forward pass, resulting in fast inference.

### 2.4  Entity Typing Score

Previous work in English EL (Raiman and Raiman, 2018; Onoe and Durrett, 2020) has shown that using an entity typing model to link entities in a KB can improve the accuracy of the EL task. Thus, we train a fine-grained entity typing model to predict entity types $t$ of each mention $\mathbf{m_i}$ where $t$ is a set of entity types $t \in T$ from the KB. We add a linear layer $f_{\theta_1}$ with a sigmoid activation $\sigma$ to map $\mathbf{m_i}$ to a fixed-size vector. We then calculate the entity typing score $\text{ET}(e_j, m_i)$ using Euclidean distance

between $\sigma(f_{\theta_1})$ and multi-label entity types $T'$:

$$\text{ET}(e_j, m_i) = ||\sigma(f_{\theta_1}(\mathbf{m_i})) - T'|| \qquad (1)$$

We formulate $T'$ by assigning a value of one to the correct entity types in $T$ and a value of zero to the rest (one entity can have multiple types). We then minimize the distance between the gold label ($T'$) and $\text{ET}(\cdot)$ using cross-entropy $\mathcal{L}_{\text{ET}}$ following the distantly-supervised type labels from Onoe and Durrett (2020).

### 2.5 Entity Description Score

In this module, we compute the cross-lingual similarity score between the mention $m_i$ and entity description ($d_j$) in the KB. We use English as the primary language of $d_j$ since English is the dominant language for mBERT (the language with the highest amount of data), and the model tends to perform substantially better compared to other languages (Arivazhagan et al., 2019; Limkonchotiwat et al., 2022). When English is unavailable we randomly select another language. We use another mBERT to encode entity descriptions $d_j$ and train jointly with our mention encoder. We formulate $d_j$ as *[CLS] label [SEP] description [SEP]* and derive the contextualized representation $\mathbf{d_j}$ from the [CLS] token in the final layer embedding. We incorporate linear layers $f_{\theta_2}$ and $f_{\theta_3}$ to mention and description encoders, respectively, with L2-normalization at the output of linear layers.

Prior works used cosine similarity to derive description score for each entity (Botha et al., 2020; FitzGerald et al., 2021; Ayoola et al., 2022). In contrast, we employ NT-Xent (the temperature-scaled cross-entropy loss) as our training objective since it demonstrated better robustness in ranking results compared to cosine similarity (Chen et al., 2020):

$$\mathcal{L}_{\text{ED}} = -\log \frac{e^{\text{sim}(f_{\theta_2}(\mathbf{m_i}), f_{\theta_3}(\mathbf{d_j}))/\tau}}{\sum_{\mathbf{d} \in \mathbf{D}} e^{\text{sim}(f_{\theta_2}(\mathbf{m_i}), f_{\theta_3}(\mathbf{d}))/\tau}} \quad (2)$$

where $\mathbf{D}$ denotes a set of description produced by candidate generation, $\tau$ denotes the temperature parameter, and $\text{sim}(\cdot)$ denotes the cosine similarity between two feature vectors.

### 2.6 Entity Disambiguation Score

Prior studies demonstrated that using entity description benefits entity disambiguation (Logeswaran et al., 2019; Wu et al., 2020). Therefore, we concatenate three outputs: (i) entity typing score (ET); (ii) the cross-lingual similarity score between

$\mathbf{m_i}$ and $\mathbf{d_j}$ with additional temperature scaling $\tau$ (similarity score calibration; Guo et al., 2017); and (iii) entity prior score ($\hat{\text{P}}(e|m)$) derived from Wikipedia hyperlink count statistics and Wikidata aliases (Hoffart et al., 2011). These three outputs are passed through a linear layer $f_{\theta_4}$ with an output dimension of one, as shown below:

$$\text{EL} = f_{\theta_4}(\text{ET}(e_j, m_i); \text{sim}(\mathbf{m_i}, \mathbf{d_j})/\tau; \hat{\text{P}}(e_j|m_i)) \tag{3}$$

The output from EL is a score for each $e_j$ corresponding to $m_i$. We train the EL score by minimizing the difference between EL and the gold label using cross-entropy $\mathcal{L}_{\text{EL}}$.

### 2.7 Multi-task training

We train mReFinED in a multi-task manner by combining all losses in a single forward pass.

$$\mathcal{L} = \lambda_1 \mathcal{L}_{\text{MD}} + \lambda_2 \mathcal{L}_{\text{ET}} + \lambda_3 \mathcal{L}_{\text{ED}} + \lambda_4 \mathcal{L}_{\text{EL}} \quad (4)$$

During the training process, we use provide entity mentions (hyperlink markups), and we simultaneously train MD (hyperlink markups and new annotations in Section 2.1) along with other tasks.

## 3 Experiments

### 3.1 Experiment Setting

**Setup**. We used the Wikipedia data and articles from 11 languages with a timestamp of 20221203 as our training data. To generate candidates, we used the top-30 candidates from Ayoola et al. (2022) and concatenated them with the top-30 candidates from Cao et al. (2022). We then select only the top 30 candidates with the highest entity prior scores for both training and inference steps. For the full parameter, language, and candidate generation settings, please refer to Appendix A.1 and A.2.

**Metric**. We evaluate mReFinED on the end-to-end EL task in both MEL datasets (Mewsli-9 and TR2016$^{\text{hard}}$) using the same metric in previous MEL works, which is based on the recall score (Botha et al., 2020; FitzGerald et al., 2021; Cao et al., 2022).

### 3.2 End-to-End MEL Results

Table 1 presents the performance of mReFinED and mGENRE on both MEL datasets. The performance of mReFinED is compared with that of a two-stage model, which involves bootstrapping MD with mGENRE. Our experiment results on Mewsli-9 demonstrated that mReFinED outperforms the

| Model | Mewsli-9 | | | | | | | | | | | TR2016[hard] | | | | |
|---|---|---|---|---|---|---|---|---|---|---|---|---|---|---|---|---|
| | ar | de | en | es | fa | ja | sr | ta | tr | macro | micro | de | es | fr | it | macro |
| MD+mGENRE | 59.0 | 67.9 | 62.1 | 67.5 | 54.0 | 38.3 | 83.7 | **34.8** | 46.4 | 57.1 | 63.6 | **30.5** | 31.5 | 23.4 | 25.5 | 27.7 |
| mReFinED | **61.8** | 69.3 | **64.2** | **68.0** | 54.2 | **43.5** | **84.5** | 33.7 | 49.8 | **58.8** | **65.5** | 28.2 | **34.4** | 25.3 | 25.8 | 28.4 |
| w/o entity priors | 60.5 | 62.5 | 61.3 | 63.5 | **54.5** | 42.8 | 83.0 | 33.2 | 47.2 | 56.5 | 61.9 | 29.1 | 34.2 | **25.8** | **26.0** | **28.8** |
| w/o entity types | **61.8** | 69.4 | **64.2** | 67.7 | 54.1 | 43.1 | 84.4 | 33.6 | 49.8 | 58.7 | 65.4 | 28.0 | 32.7 | 24.5 | 25.5 | 27.7 |
| w/o descriptions | **61.8** | **69.9** | 63.0 | 67.2 | 53.2 | 42.4 | 84.0 | 33.5 | **50.1** | 58.3 | 65.0 | 8.2 | 17.8 | 9.6 | 9.6 | 11.3 |
| w/o bootstrapping | 0.0 | 0.1 | 0.0 | 0.0 | 0.0 | 0.0 | 0.0 | 0.0 | 0.0 | 0.0 | 3.0 | 0.0 | 0.0 | 0.0 | 0.0 | 0.0 |

Table 1: Recall on Mewsli-9 and TR2016[hard] datasets. We report both datasets' results of entity disambiguation and entity linking tasks. We use the bootstrapping MD model as mention detection for mGENRE.

two-stage model on the micro- and macro-averages by 1.8 and 1.9 points, respectively. The experimental results from TR2016[hard] also demonstrated the same results as Mewsli-9. These results highlight the essentials of the end-to-end system in a single model, which is better than the two-stage model. For precision and F1 scores, see table 8 in Appendix. The ablations in Table 1 show that, when the bootstrapping MD framework is removed and using only the Wikipedia markups, mReFinED produces zero scores for almost all languages in both datasets. This is because the number of entity mentions in the training data decreased from 880 million to only 180 million mentions. These results emphasize the importance of our bootstrapping MD framework, which can effectively mitigate the unlabelled entity mention problem in Wikipedia.

On Mewsli-9, entity priors and descriptions are slightly complementary and contribute +0.5 macro and micro average recall when combined. Entity types are less useful and contribute +0.1 on macro and micro average recall when added. Combining all three achieves the best macro and micro average recall - 58.8 and 65.5 respectively.

For Arabic (ar), removing either entity types or descriptions has no difference and achieves the same recall (61.8) compared to using all information. This suggests entity types and descriptions are redundant when either one of them is combined with entity priors. Entity priors hurt the performance in Farsi/Persian (fa) as removing it gives +0.3 recall. However, there are only 535 Farsi mentions in the Mewsli-9 dataset, which is too small of a sample size to draw reliable conclusions. For German (de) and Turkish (tr), removing descriptions seems to be beneficial and yields recall gains of +0.6 and +0.3 respectively. This could be a resource-specific issue (there could be fewer/lower quality Wikidata descriptions in these two languages) or a language-related issue (both languages are morphologically rich) but we will

leave further investigations for future work.

On TR2016[hard], entity descriptions show the most contribution, +17.1 macro average recall when added. Entity types show a small amount of contribution, with +0.7 macro average recall. Entity priors turn out to be harmful when added except for es language. Macro average recall is +0.4 when entity priors are removed. This could be explained by how mentions in TR2016[hard] dataset are selected. Mentions in TR2016[hard] dataset are chosen so that the correct entity did not appear as the top-ranked candidate by alias table lookup. This means entity priors are not very useful for finding the correct entity for these mentions and model needs to use other information such as entity descriptions and types to choose the correct entity. We believe this introduces a discrepancy to the training scenario where entity priors are very useful signal for finding the correct entity given a mention surface form. On the other hand, since the gap between with and without entity priors is small, it also demonstrates mReFinED model's ability to use appropriate information when entity priors alone is not enough to make correct predictions.

### 3.3 Multilingual Mention Detection Results

This experiment compares the performance of our mention detection models with prior multilingual MD works, such as spaCy (Honnibal et al., 2020), XTREME (Hu et al., 2020), and WikiNEuRal (Tedeschi et al., 2021) on both MEL datasets. We use the exact match score to evaluate the efficiency of this study following previous MD works (Tjong Kim Sang and De Meulder, 2003; Tsai et al., 2006; Diab et al., 2013). As shown in Table 9, our bootstrapping MD outperforms competitive methods in all languages; e.g., our bootstrapping MD outperformed XTREME by 9.9 points and 7.3 points on Mewsli-9 and TR2016[hard] datasets, respectively. In addition, mReFinED showed superior performance to the bootstrapping

MD by an average of 2.8 points on Mewsli-9. These results highlight the benefits from additional join training MD with other tasks in a single model outperformed a single task model. We also run an experiment on XTREME NER dataset to better understand our bootstrapping MD's performance on multilingual mention detection task. We expect our bootstrapping MD to achieve comparable results to competitive multilingual NER models in the literature when trained on NER data, please refer to Appendix A.5 for more details.

### 3.4 Analysis

**Incorrect labels in MEL datasets**. It is noteworthy that both Mewsli-9 and TR2016 datasets contains incorrect labels. In particular, we identified entities that were erroneously linked to the "Disambiguation page" instead of their actual pages; e.g., the mention "imagine" in Mewsli-9 was linked to Q225777 – a "Wikimedia disambiguation page". Therefore, we removed those incorrect labels in both datasets and re-evaluated mReFinED and mGENRE on the cleaned dataset in Table 2. mReFinED's performance on the cleaned Mewsli-9 dataset increases from 65.5 to 67.4 micro-avg, and mGENRE's performance increases from 63.6 to 65.7. Lastly, the number of entity mention in Mewsli-9 was decreased from 289,087 to 279,428 mentions. See Appendix A.4 for TR2016 results.

| Method | Cleansed Mewsli-9 | | | | | | | | | | |
|---|---|---|---|---|---|---|---|---|---|---|---|
| | ar | de | en | es | fa | ja | sr | ta | tr | macro | micro |
| MD+mGENRE | 60.3 | 69.9 | 64.7 | 70.0 | 54.4 | 39.6 | 85.4 | **34.9** | 47.6 | 58.5 | 65.7 |
| mReFinED | **63.0** | **71.3** | **66.8** | **70.1** | **54.6** | **44.0** | **86.3** | 34.0 | **50.9** | **60.1** | **67.4** |

Table 2: Recall score on the cleaned Mewsli-9 dataset.

**Unlabelled entity mentions in MEL datasets**. It is important to note that the unlabelled entity mention problem also occurs in both MEL datasets. As mentioned in Section 2.3, most of entities in MEL datasets are common noun because these datasets use Wikipedia markups as entity mention ground truths. Thus, the MEL datasets also suffer from the unlabelled entity mention problem. For example, consider the document en-106602 in Mewsli-9 (Figure 3), it was annotated with only eight entity mentions, but mReFinED found an additional 11 entity mentions in the document, including location (i.e., "Mexico"), person (i.e., "Richard A. Feely"), organization (i.e., "NOAA"), and common noun (i.e., "marine algae") mentions. These results demonstrate that mReFinED can also mitigate the unlabelled entity mention in MEL datasets.

This presents an opportunity for us to re-annotate the MEL datasets in the future using mReFinED as an annotator tool to detect unlabelled mentions.

**Run-time Efficiency**. This study measures the time per query of mReFinED compared to mGENRE on one 16 GB V100. Our findings indicate that mGENRE takes 1,280 ms ± 36.1 ms to finish a single query. In contrast, mReFinED requires only 29 ms ± 1.3 ms making it 44 times faster than mGENRE because mReFinED encodes all mentions in a single forward pass.

## 4 Conclusion

In this paper, we propose mReFinED, the first multilingual end-to-end EL. We extend the monolingual ReFinED to multilingual and add the new bootstrapping MD framework to mitigate the unlabelled mention problem. mReFinED outperformed SOTA MEL in the end-to-end EL task, and the inference speed is faster than SOTA 44 times.

## Limitations

We did not compare mReFinED with other MEL works (Botha et al., 2020; FitzGerald et al., 2021) since they did not release their code. However, the experimental results from other MEL works in the ED task demonstrated lower performance than mGENRE. Our experiment report results are based on standard MEL datasets, such as Mewsli-9 and TR2016hard, which may not reflect mReFinED's performance in real-world applications.

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

## A    Appendix

### A.1    Setup

We trained our model on 11 languages: ar, de, en, es, fa, fr, it, ja, ta, and tr. During training, we segment the training document into chucks consisting of 300 tokens each, and limited the mention per chunk only 30 mentions. We use two layers of mBERT as the description encoder. We train mReFinED for eight days using the AdamW optimizer, with a learning rate of $5e^{-4}$, and a batch size of 64 for two epochs on 8 A100 (40 GB). For hyper-parameter settings, we set the hyper-parameters as shown in Table 3. In addition, we evaluate the recall score of the development set in every 2,000 steps to save the best model.

| Hyper-parameter | Value |
|---|---|
| $\lambda_1$ | 0.01 |
| $\lambda_2$ | 1 |
| $\lambda_3$ | 0.01 |
| $\lambda_4$ | 1 |
| $\tau$ | 0.02 |

Table 3: Hyper-parameters of mReFinED.

### A.2    Candidate Generation (CG) Results

In this experiment, we demonstrated the recall score of various candidate generation methods on both MEL datasets and why we need to combine two CG. We adopt ReFinED's CG from monolingual to multilingual PEM tables using multilingual Wikipedia data and articles. As shown in Table 4, mGENRE's CG outperformed ReFinED's CG by 3.4 points on the average case. This is because mGENRE's CG was formulated from Wikipedia in 2019, while ReFinED's CG was formulated in 2022, and there are many rare candidates that are not found in ReFinED's CG but appear in mGENRE's CG. On the other hand, Table 5 demonstrates that ReFinED's CG outperformed mGENRE's CG on the TR2016[hard] dataset. Thus, combining two CGs outperforms using only one CG on both MEL datasets.

### A.3    Bootstrapping MD Results

In this study, we demonstrated the effect of training size on bootstrapping MD framework. For the training data sizes, we set the size as follows: 10k, 20k, 100k, 200k, 500k, 1M. As shown in Figure 2, the training data has affected the performance of the bootstrapping MD framework. However, we

| Language | Ayoola et al. (2022) | Cao et al. (2022) | mReFinED |
|---|---|---|---|
| ar | 91.0 | 95.8 | **96.6** |
| de | 89.9 | 93.9 | **95.7** |
| en | 86.3 | 95.9 | **96.7** |
| es | 90.1 | 89.6 | **94.4** |
| fa | 90.7 | 90.1 | **92.3** |
| ja | 90.5 | 90.1 | **91.5** |
| sr | 91.4 | 96.1 | **97.2** |
| ta | 82.4 | 88.4 | **92.9** |
| tr | 88.7 | 91.4 | **95.1** |
| **macro-avg** | 89.0 | 92.4 | **94.7** |

Table 4: Candidate generation score (recall) on Mewsli-9. We use only top-30 candidates.

| Language | Ayoola et al. (2022) | Cao et al. (2022) | mReFinED |
|---|---|---|---|
| de | 84.9 | 74.1 | **85.3** |
| es | 83.2 | 67.0 | **87.0** |
| fr | 81.9 | 59.6 | **83.0** |
| it | 79.6 | 76.7 | 84.6 |
| **macro-avg** | 83.7 | 69.4 | **85.0** |

Table 5: Candidate generation score (recall) on TR2016[hard]. We use only top-30 candidates.

found that increasing the number of training data more than 200k samples does not increase the performance of MD significantly.

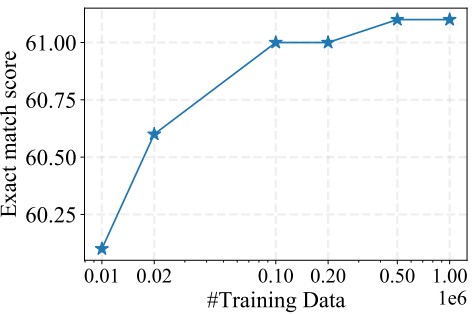

Figure 2: Effect of training size in the bootstrapping MD framework.

### A.4    Incorrect Labels TR2016 Results

This study demonstrates the performance of mGENRE and mReFinED on the cleaned TR2016[hard] dataset. As shown in Table 6, we observe a 1.5 points improvements from the original result 1 (28.4). Lastly, the number of entity mentions was decreased from 16,357 to 15,380.

### A.5    XTREME MD Results

To evaluate the performance of mention detection in mReFinED, we run an experiment on XTREME NER dataset by converting the labels to mention detection task with BIO tags. We chose a subset of 8 languages from XTREME NER dataset as our Bootstrapping MD is trained on 9 languages in

| Cleansed TR2016hard | | |
|---|---|---|
| **Language** | **MD+mGENRE** | **mReFiNED** |
| de | **31.0** | 29.1 |
| es | 33.1 | **36.8** |
| fr | 26.6 | **26.7** |
| it | 26.3 | **27.1** |
| **macro-avg** | 29.3 | **29.9** |

Table 6: Recall score on the cleansed TR2016hard dataset (removed entities that linked to disambiguation pages).

Mewsli-9 dataset. Language sr is dropped here because it's not available in XTREME NER dataset. Results of F1 scores are shown in table 7. Please note that F1 scores of Hu et al. (2020) are for NER task with three entity types - LOC, PER and ORG. The fine-tuned models used to produce results in their paper are not released so we couldn't produce the results for mention detection task. But we want to include their NER results here to better understand our Bootstrapping MD's performance on multilingual mention detection task and we expect to achieve comparable results to competitive multilingual NER models in the literature if trained on NER data.

| Model | XTREME | | | | | | | | |
|---|---|---|---|---|---|---|---|---|---|
| | ar | de | en | es | fa | ja | ta | tr | macro |
| Hu et al. (2020) | 53.0 | **78.8** | **85.2** | **79.6** | 61.9 | 31.2 | 59.5 | 76.1 | 65.7 |
| Bootstrapping MD | **69.2** | 74.6 | 73.6 | 78.0 | **71.8** | **44.5** | **72.9** | **81.4** | **70.7** |

Table 7: F1 score of mention detection on XTREME NER dataset.

## A.6 GenBench Evaluation Card

| Motivation | | | |
|---|---|---|---|
| *Practical* ☐ | *Cognitive* | *Intrinsic* | *Fairness* |
| **Generalisation type** | | | | | |
| *Compositional* | *Structural* | *Cross Task* | *Cross Language* ☐ | *Cross Domain* ☐ | *Robustness* |
| **Shift type** | | | |
| *Covariate* | *Label* | *Full* | *Assumed* ☐ |
| **Shift source** | | | |
| *Naturally occuring* ☐ | *Partitioned natural* ☐ | *Generated shift* | *Fully generated* |
| **Shift locus** | | | |
| *Train–test* | *Finetune train–test* ☐ | *Pretrain–train* | *Pretrain–test* |

| Model | Mewsli-9 | | | | | | | | | | | TR2016$^{hard}$ | | | | |
|---|---|---|---|---|---|---|---|---|---|---|---|---|---|---|---|---|
| | ar | de | en | es | fa | ja | sr | ta | tr | macro | micro | de | es | fr | it | macro |
| MD+mGENRE Precision | 11.3 | 18.7 | 15.9 | 17.7 | 12.4 | 20.8 | 12.9 | 5.7 | 14.8 | 14.5 | 16.2 | 1.4 | 1.2 | 1.1 | 1.3 | 1.2 |
| MD+mGENRE Recall | 59.0 | 67.9 | 62.1 | 67.5 | 54.0 | 38.3 | 83.7 | **34.8** | 46.4 | 57.1 | 63.6 | **30.5** | 31.5 | 23.4 | 25.5 | 27.7 |
| MD+mGENRE F1 | 19.0 | 29.4 | 25.4 | 28.1 | 20.2 | 27.0 | 22.3 | 9.8 | 22.5 | 22.6 | 25.8 | 2.7 | 2.3 | 2.0 | 2.5 | 2.4 |
| mReFinED Precision | **14.9** | **21.1** | **18.8** | **21.0** | **15.5** | **28.4** | **17.0** | **9.7** | **21.0** | **18.6** | **19.7** | **1.6** | **1.7** | **1.5** | **1.7** | **1.6** |
| mReFinED Recall | **61.8** | **69.3** | **64.2** | **68.0** | **54.2** | **43.5** | **84.5** | 33.7 | **49.8** | **58.8** | **65.5** | 28.2 | **34.4** | **25.3** | **25.8** | **28.4** |
| mReFinED F1 | **24.0** | **32.3** | **29.0** | **32.1** | **24.1** | **34.3** | **28.3** | **15.0** | **29.6** | **27.6** | **30.3** | **3.1** | **3.2** | **2.9** | **3.2** | **3.1** |

Table 8: Precision, Recall and F1 scores on Mewsli-9 and TR2016$^{hard}$ datasets. We report both datasets' results of entity disambiguation and entity linking tasks. We use the bootstrapping MD model as mention detection for mGENRE. The low precision and F1 scores are due to unlabelled entity mentions in MEL datasets as discussed in Section 3.4

| Model | Mewsli-9 | | | | | | | | | | TR2016 | | | | |
|---|---|---|---|---|---|---|---|---|---|---|---|---|---|---|---|
| | ar | de | en | es | fa | ja | sr | ta | tr | macro | de | es | fr | it | macro |
| SpaCy | - | 50.6 | 56.0 | 59.3 | - | - | 21.3 | 4.1 | 32.7 | - | 42.0 | 53.4 | 42.1 | 44.3 | 45.5 |
| XTREME | 49.8 | 71.7 | 63.4 | 70.6 | 40.6 | 12.6 | 80.3 | 24.4 | 46.4 | 51.1 | 43.3 | 51.0 | 40.9 | 46.9 | 45.5 |
| WikiNEuRal | 52.0 | 72.2 | 68.9 | 75.1 | - | - | - | - | - | - | 37.6 | 56.3 | 42.4 | 46.6 | 45.7 |
| Bootstrapping | 60.7 | 72.2 | 69.5 | 73.9 | 55.5 | 43.6 | 87.3 | **36.6** | 49.9 | 61.0 | **53.3** | 58.4 | **48.0** | **51.3** | **52.8** |
| mReFinED | **63.5** | **75.5** | **71.7** | **75.8** | **57.2** | **47.7** | **89.4** | 36.4 | **56.7** | **63.8** | 51.7 | **59.0** | 47.6 | 50.4 | 52.4 |

Table 9: Exact match score of mention detection on on Mewsli-9 and TR2016$^{hard}$ datasets. We omitted unsupported languages for each model with "-".

Figure 3: The unlabelled entity mention samples from Section 3.4