# OpenReview forum: "mReFinED: An Efficient End-to-End Multilingual Entity Linking System"
_EMNLP/2023/Conference — EMNLP 2023 Findings_

### Official Review · Reviewer_9qjt · 2023-07-30

**Soundness:** 4

**Excitement:**

4: Strong: This paper deepens the understanding of some phenomenon or lowers the barriers to an existing research direction.

**Paper Topic And Main Contributions:**

This paper proposes the first technique for creating an end-to-end multilingual entity linker (MEL), mReFinED based on monolingual version ReFinED. Prior MELs only attempted the linking started with the from identified mentions. A key part to getting ReFinED to succeed at the task was to create a bootstrapped dataset to annotate unlabeled mentions in Wikipedia. The approach was evaluated on two datasets, Mewsli-9 and TR2016^hard. The end-to-model outperforms two-stage models on both datasets.

**Reasons To Accept:**

A major strength of this paper is proposing the first end-to-end MEL system that generally performs better than the two-stage approach in terms of recall in most languages. The description of the approach is clear, and there is sufficient detail to reproduce the proposed method of bootstrapping strapping.

**Reasons To Reject:**

One of the main weaknesses of the paper is that neither F1 or precision are looked at in either the analysis or appendix. I understand following prior work to report only recall, but for a system to be usable it must also perform well on a precision-oriented metric. Given the lack of analysis in terms of precision, it makes me wonder how usable the mReFinED model is.

A second weakness is the choice to analyze the dataset rather than analyze the algorithm in Section 3.4. Evaluation data does not need to be perfect as long as it is not systematically biased against or in favor on any one particular system since such errors will depress the scores of all systems that utilize the dataset. It is important to remember that evaluation is about determining relative performance of systems rather than absolute values. I would much prefer such dataset analysis in the appendix, and to instead include a qualitative analysis of the algorithm.


**Reproducibility:**

4: Could mostly reproduce the results, but there may be some variation because of sample variance or minor variations in their interpretation of the protocol or method.

**Reviewer Confidence:**

3: Pretty sure, but there's a chance I missed something. Although I have a good feel for this area in general, I did not carefully check the paper's details, e.g., the math, experimental design, or novelty.

**Typos Grammar Style And Presentation Improvements:**

Mention how you handle overlapping entities in Section 2

Table 2: Is MD missing from mGENRE?
L347 codes -> code

---

> ### Author Rebuttal · Authors · 2023-08-28
>
> (1) One of the main weaknesses of the paper is that neither F1 or precision are looked at in either the analysis or appendix. I understand following prior work to report only recall, but for a system to be usable it must also perform well on a precision-oriented metric. Given the lack of analysis in terms of precision, it makes me wonder how usable the mReFinED model is.
>
> We thank you for the valuable suggestion. As stated in section 3.4, both Mewsli-9 and TR-2016 datasets suffer unlabelled entity mention problem, this will penalise precision despite correct entity mention and linking predictions are made, there is an example in figure 1 in Appendix. For the revised version, we will include F1 and precision scores.
>
> (ii) A second weakness is the choice to analyze the dataset rather than analyze the algorithm in Section 3.4. Evaluation data does not need to be perfect as long as it is not systematically biased against or in favor on any one particular system since such errors will depress the scores of all systems that utilize the dataset. It is important to remember that evaluation is about determining relative performance of systems rather than absolute values. I would much prefer such dataset analysis in the appendix, and to instead include a qualitative analysis of the algorithm.
>
> For the design choice of our framework, we carefully choose each component to achieve SOTA based on the study of ReFinED, such as the design analysis of entity prior, entity type, and description training objectives. In the camera-ready version, we will include this experiment in Section 3.5.
>
> (3) Typos Grammar Style And Presentation Improvements
>
> In the revised version, we will address all comments, including how we handle overlapping entities.

---

### Official Review · Reviewer_4Pgx · 2023-07-31

**Soundness:** 3

**Excitement:**

3: Ambivalent: It has merits (e.g., it reports state-of-the-art results, the idea is nice), but there are key weaknesses (e.g., it describes incremental work), and it can significantly benefit from another round of revision. However, I won't object to accepting it if my co-reviewers champion it.

**Paper Topic And Main Contributions:**

The text explores the concept of end-to-end multilingual entity linking (MEL), a procedure that pinpoints multilingual entity mentions and associates them with the relevant entity IDs within a knowledge base. Current approaches circumvent the entity mention detection stage, attributable to the scarcity of high-quality multilingual training datasets. To rectify this shortfall, the authors introduce mReFinED, the inaugural comprehensive MEL system. They further propose an innovative bootstrapping mention detection framework designed to enhance the caliber of training corpora. Empirical data illustrates that mReFinED outshines its predecessors in MEL assignments, while simultaneously offering an efficiency improvement of 44 times.

**Reasons To Accept:**

(1) This paper pioneers the introduction of the first-ever end-to-end multilingual entity linking (MEL) within a unified model, effectively expanding the reach of ReFinED into multilingual ReFinED. This represents a consequential leap forward within the discipline, providing a holistic solution for MEL.
(2) The authors augment their contribution by suggesting a novel bootstrapping mention detection framework. This ingenious methodology rectifies the widespread challenge of unlabelled mentions in end-to-end MEL, amplifying both the precision and speed of entity linking processes.

**Reasons To Reject:**

(1) The paper fails to offer a comparative analysis with a selection of mainstream large language models currently available, such as ChatGPT, GPT-4, and LLAMA. This omission hampers a well-rounded performance assessment of the proposed methods.
(2) The predominance of zero values in the 'w/o bootstrapping' test results is troubling. The vast discrepancy between these results and the ME test results fails to convincingly demonstrate the efficacy of bootstrapping method, leading me to question the validity of these tests. I recommend that more extensive testing be carried out across a broader range of datasets to better validate the benefits of the bootstrapping approach.

**Reproducibility:**

4: Could mostly reproduce the results, but there may be some variation because of sample variance or minor variations in their interpretation of the protocol or method.

**Reviewer Confidence:**

4: Quite sure. I tried to check the important points carefully. It's unlikely, though conceivable, that I missed something that should affect my ratings.

---

> ### Author Rebuttal · Authors · 2023-08-28
>
> (1) The paper fails to offer a comparative analysis with a selection of mainstream large language models currently available, such as ChatGPT, GPT-4, and LLAMA. This omission hampers a well-rounded performance assessment of the proposed methods.
>
> We thank you for the suggestion. For the LLMs experiment, we think it is not realistic to expect a LLM to store an entire KG for candidate retrieval unless backed by an API, we also agree with the opinions of Anna Rogers and other in the NLP community that evaluating against closed LLMs is not ideal for various reasons (see https://hackingsemantics.xyz/2023/closed-baselines/)
>
>  (2) The predominance of zero values in the 'w/o bootstrapping' test results is troubling. The vast discrepancy between these results and the ME test results fails to convincingly demonstrate the efficacy of bootstrapping method, leading me to question the validity of these tests. I recommend that more extensive testing be carried out across a broader range of datasets to better validate the benefits of the bootstrapping approach.
>
> As stated in Section 3.2, in the ‘w/o bootstrapping’ setting, we omitted the bootstrapped mentions in our training data to show that, without the additional mentions, the model was confused to detect mention spans since there are unlabeled mention spans in the training data (as shown in Figure 3). In addition, we ran an additional mention detection experiment on XTREME NER benchmark datasets and found that the performance of our MD and SOTA is comparable. We will include this new experiment in the camera-ready version.

---

### Official Review · Reviewer_N6KL · 2023-08-03

**Soundness:** 3

**Excitement:**

3: Ambivalent: It has merits (e.g., it reports state-of-the-art results, the idea is nice), but there are key weaknesses (e.g., it describes incremental work), and it can significantly benefit from another round of revision. However, I won't object to accepting it if my co-reviewers champion it.

**Paper Topic And Main Contributions:**

In this paper, the authors proposed a framework for end-to-end multilingual entity linking (MEL). The framework, named mREFinED, consists of a bootstrapping mention detection, an entity typing, an entity description, an entity disambiguation, and combine all in a multi-task training objective. Experiments showed that mREFinED outperformed previous works, namely SpaCy, XTREME, and WikiNEuRal.
Is there any comparison between the bootstrapped MEL model and the original NER model? At least on only the original NER entity types.
The term "end-to-end" is very confusing, as far as I understood, almost every previous work also identify mentions, either using existing NER or Wikipedia markups as labelled data. So why does it claim to be the first "end-to-end"?

**Questions For The Authors:**

Is there any comparison between the bootstrapped MEL model and the original NER model? At least on only the original NER entity types.
The term "end-to-end" is very confusing, as far as I understood, almost every previous work also identify mentions, either using existing NER or Wikipedia markups as labelled data. So why does it claim to be the first "end-to-end"?

**Reasons To Accept:**

Experiments showed that mREFinED outperformed previous works, namely SpaCy, XTREME, and WikiNEuRal.

**Reasons To Reject:**

Is there any comparison between the bootstrapped MEL model and the original NER model? At least on only the original NER entity types.
The term "end-to-end" is very confusing, as far as I understood, almost every previous work also identify mentions, either using existing NER or Wikipedia markups as labelled data. So why does it claim to be the first "end-to-end"?

**Reproducibility:**

4: Could mostly reproduce the results, but there may be some variation because of sample variance or minor variations in their interpretation of the protocol or method.

**Reviewer Confidence:**

4: Quite sure. I tried to check the important points carefully. It's unlikely, though conceivable, that I missed something that should affect my ratings.

---

> ### Author Rebuttal · Authors · 2023-08-28
>
> (1) Is there any comparison between the bootstrapped MEL model and the original NER model? At least on only the original NER entity types.
>
> We thank you for the suggestion. The result of entity mention detection model without bootstrapping is similar to the result of entity linking without bootstrapping in table 1 so we didn’t include it in table 7 in Appendix where we compare our entity mention model with bootstrapping to SpaCY, XTREME and WikiNEuRal. Our work focuses on entity linking not NER so we trained entity mention detection model which doesn’t further distinguish among various entity types, we will add a row in table 7 to show result of entity mention detection model without bootstrapping in revised version.
>
> (2) The term "end-to-end" is very confusing, as far as I understood, almost every previous work also identify mentions, either using existing NER or Wikipedia markups as labelled data. So why does it claim to be the first "end-to-end"?
>
> Our work focuses on entity linking not NER, it is true that previous MEL work referenced in our paper use Wikipedia markups as labelled data in their training, but entity mentions were assumed given at inference time along with raw text input. We don’t consider previous MEL work end-to-end because of this assumption. Our work doesn’t make this assumption and identifies both entity mention and id given just raw text input hence the claim of the first end-to-end MEL.

---

### Meta-Review · Area_Chair_5ccu · 2023-09-24

**Recommendation:** 4

**Metareview:**

This paper presents a study that is able to perform end-to-end multilingual entity linking. The reviewers are generally positive on this work, especially given it's a first end-to-end approach to such a problem. One concern from the reviewers is that the evaluation can be further strengthened (e.g., with more evaluation metrics). Overall this is a good short paper that contains interesting research progress that is worth sharing with the community.

---

### Decision · Program_Chairs · 2023-10-07

**Decision:**

Accept-Findings

**Comment:**

This paper presents a study that is able to perform end-to-end multilingual entity linking. The reviewers are generally positive on this work, especially given it's a first end-to-end approach to such a problem. One concern from the reviewers is that the evaluation can be further strengthened (e.g., with more evaluation metrics). Overall this is a good short paper that contains interesting research progress that is worth sharing with the community.